# CVX-DPO: Fast Resource Constrained Preference Learning via Convex Optimization

## Abstract

Fine-tuning large language models (LLMs) for alignment with human preferences have become a key factor in the success of models like ChatGPT and Gemini, which are now integral to mainstream use. Many effective techniques are based on Reinforcement Learning from Human Feedback (RLHF), yet are challenging and expensive to implement. Direct Preference Optimization (DPO) offers an accessible alternative by simplifying the objective, but can exhibit random ranking accuracy and requires a frozen reference model. In this paper, we develop a fast and an even more lightweight DPO based algorithm — *CVX-DPO* —- that operates on a single GPU. The key to achieving this is leveraging the convex optimization reformulation of neural networks, which eliminates the dependence on copying the reference model and is robust against hyperparameter tuning. CVX-DPO can be trained to global optimality in polynomial time. We use the Alternating Direction Method of Multipliers (ADMM) to solve this optimization problem in order to increase parallelization efficiency, and implement our methods in JAX to lift the memory constraints across experiments. We experiment on three datasets, including one synthetically generated educational dataset, to demonstrate the efficacy of our novel algorithm in a real world setting. CVX-DPO outperforms traditional DPO in user preference generation when tested on human subjects, despite being trained on one single RTX-4090 GPU.

## 1 Introduction

Language models have been trained on increasingly large amounts of data to capture semantic language patterns. The current paradigm is a combination of pre-training and fine-tuning these LMs to achieve aligned user preferable responses. Reinforcement learning from human feedback (RLHF) (Stiennon et al., 2020; Ouyang et al., 2022; Christiano et al., 2017; Wang et al., 2023) has demonstrated impressive results to achieve alignment, and utilizes a three step approach of: supervised fine-tuning, reward model training, and policy optimization. However this complexity presents several optimization and resource challenges in its multi-stage approach. Recently, the DPO (Rafailov et al., 2024) algorithm proposes a simpler and more computationally lightweight alternative to aligning LMs for user preferred responses. DPO reparametrizes the reward function instead of learning an explicit reward model and incorporates this into the Bradley-Terry ranking objective (Bradley & Terry, 1952). Although simpler, this also yields the following drawbacks: requiring a reference model to stabilize training incurs additional memory and computational costs, there exists a mismatch between the reward optimized in training and the log-likihood optimized during inference. Recent work (Chen et al., 2024; Meng et al., 2024) have shown that models trained with DPO exhibit random ranking accuracy even after extensive training. The SimPO (Meng et al., 2024) algorithm employs an implicit reward formulation that directly aligns with the generation metric, eliminating the need for a reference model yielding a more effective yet memory efficient approach. However this also introduces additional hyperparameters, and the authors note that the strategy is crucially dependent on extensive hyperparameter tuning to achieve optimal performance.

In this paper, we introduce CVX-DPO, a novel fast and lightweight framework for preference fine-tuning small language models. Our approach is based on stacking a *convex* two-layer neural network classifier on top of a pre-trained model. Moreover, we propose a new objective based on fine-tuning the convex network's last layer. Unlike the original DPO objective, this new objective is a much smaller *convex* optimization problem that can be trained to global optimality. As the convex

network and the fine-tuning objective are both convex, they can be solved using algorithms for convex optimization in a scalable and near-hyperparameter-free manner. In particular, to train the convex network, we combine the ADMM algorithm (Boyd et al., 2011) with GPU acceleration to facilitate fast training. In addition, we leverage JAX (Bradbury et al., 2018) and its Just-In-Time-Compilation (JIT) feature to lower our CVX-DPO code to fast machine code. As a result, using only one RTX-4090, CVX-DPO can fine-tune GPT-2 medium within a few minutes. Indeed, all of the experiments in this work were performed on a single RTX-4090 GPU. Despite the hardware limitation, CVX-DPO enjoys fast train times — less than one hour and, in some instances, only several minutes.

Thanks to its lightweight computational and memory footprint, CVX-DPO provides an important step forward in developing efficient ways of aligning LLMs to human preferences that are more accessible in academic settings. This helps to further democratize AI systems for wider audiences and improve optimization techniques in this area. In order to assess the efficacy of our method, we create a synthetic Educational-Tutor conversational dataset, and evaluate on 25 human volunteers via survey to understand "preference".

**Contributions.**    Our main contributions can be summarized as follows:

- We introduce a novel CVX-DPO algorithm in section 4 which reformulates the traditional non-convex DPO loss for more stable training.
- Our theoretical convergence guarantees in section 4.3 prove that we can train CVX-DPO to global optimality in polynomial time.
- CVX-DPO offers faster convergence and is extremely VRAM efficient. Our algorithm also mitigates the crucial dependence on tuning hyperparameters for achieving optimal performance exhibited in methods such as DPO and SimPO. These results are validated with extensive experiments and human evaluation.
- We develop a custom conversational dataset to simulate a real world setting, which features diverse topics and varying turns of phrase.
- Our open source JAX code base is provided for further experimentation and research, including methods for both Pytorch (Paszke et al., 2019) and FLAX (Kidger & Garcia, 2021) models.

## 2    RELATED WORK

Fine-tuning large language models (LLMs) that better align to human preferences can be approached through three distinct strategies. Initial algorithms of zero-shot and few-shot in-context learning (Xian et al., 2017) relies on prompt engineering. Although this method is able to improve the performance of LLMs to produce desired outputs and does not require fine-tuning, it is not able to tackle complex tasks. More sophisticated learning methods use reinforcement learning to align model outputs with user preferences. The most successful classes (such as RLHF and RLAIF (Lee et al., 2023)) have been able to create conversational LLMs such as ChatGPT. However despite their impressive performance, these methods are extremely complex, requires humans in the loop, and requires significant computational resources. Therefore the authors of DPO developed a simple yet performant learning algorithm to directly optimize to human preferences, without explicit reward modeling. The official implementation of DPO references four 80GB A100s, which reduces the barrier to training LLMs. Nevertheless, multi-GPU systems are still out of reach for many researchers in academia. In addition, DPO introduces certain hyperparameters that need careful tuning.

Bengio et al. (2005) have previously shown that it is possible to characterize the optimization problem for neural networks as a convex program. Pilanci & Ergen (2020) further developed exact convex reformulations of training a two-layer ReLU neural network. The core of this representation lies in semi-infinite duality theory, and was derived in **?** to show that two-layer neural networks with ReLU activations and weight decay regularization may be re-expressed as a linear model with a group one penalty and polyhedral cone constraints. This is a step towards achieving globally optimal networks and interpretable results. This yields both practical benefits in implementation, and theoretical advantages in analyzing the optimization of the non-convex landscape of NNs. This framework is most efficient on two-layer NNs, and on small scale datasets such as CIFAR-10 or

MNIST (Mu & Gilmer (2019)). In order to apply this method to the area of LLMs where large data is paramount, we seek better solutions for scalability.

To practically solve this convex optimization problem, Bai et al. (2018) have proposed an approach based on the Alternating Direction Method of Multipliers (ADMM) (Boyd et al., 2011). ADMM offers several attractive advantages, such as its robustness against hyperparameter selection, linear decomposability for distributed optimization, and immunity to vanishing/exploding gradients. The successful application of ADMM in solving optimization problems across a wide range of domains has been well studied. This includes diverse fields such as control theory (Li et al. (2017)), maximum a posteriori (MAP) inference problems (Lu & Lü (2019)), computational biology and finance (Costa & Kwon (2020)). The natural parallelization aspects of ADMM seem to make it particularly suitable to deep learning problems. Therefore we aim to integrate the convex reformulations of NNs with DPO. Our goal is to have the convex model provide clear signals to the DPO loss, thus leveraging the faster convergence to obtain LLMs of better quality.

## 3 CONVEX NEURAL NETWORKS

In this section, we review convex neural networks, which sets the stage for our introduction of the Convex DPO (CVX-DPO) algorithm.

### 3.1 TWO-LAYER ReLU NETWORKS

Given an input $x \in \mathbb{R}^d$, the classic two-layer ReLU network is given by:

$$f(x) = \sum_{j=1}^{m} (\Theta_{1j}x)_+ \theta_{2j}, \tag{1}$$

where $\Theta_1 \in \mathbb{R}^{m \times d}$, $\theta_2 \in \mathbb{R}^m$ are weights of the first and last layer respectively, and $(\cdot)_+ = \max\{\cdot, 0\}$ is the ReLU activation function.

Given targets $y \in \mathbb{R}^n$, the network in equation 1 is trained by minimizing the following non-convex loss function:

$$\min_{\Theta_1, \theta_2} \ell\left(f_{\Theta_1, \theta_2}(X), y\right) + \frac{\beta}{2} \sum_{j=1}^{m} \left(||\Theta_{1j}||_2^2 + (\theta_{2j})^2\right), \tag{2}$$

where $\ell : \mathbb{R}^n \mapsto \mathbb{R}$ is the loss function, $X \in \mathbb{R}^{n \times d}$ is the data matrix, and $\beta \geq 0$ is the regularization strength. Solving equation 2 is challenging due to the non-convexity of the objective. The optimizer often needs meticulous tuning of hyperparameters to ensure successful training. Such tuning is expensive, since it requires many iterations of running the optimizer across multiple hyperparameter configurations in a grid search to obtain good performance. This dramatically contrasts with the convex optimization framework, where algorithms come with strong convergence guarantees and involve minimal hyperparameters. Fortunately, it is possible to maintain the expressive capabilities of ReLU neural networks while still enjoying the computational advantages of convex optimization.

### 3.2 CONVEX REFORMULATION

Pilanci & Ergen (2020) have shown equation 2 admits a convex reformulation, which makes it possible to avoid the inherent difficulties of non-convex optimization. Significantly, the reformulation has the same optimal value as the original non-convex problem, provided $m \geq m^*$, for some $m \geq n+1$. Therefore, nothing is lost in reformulating equation 2.

Pilanci & Ergen (2020)'s convex reformulation is based on enumerating the actions of all possible ReLU activation patterns on the data matrix $X$. These activation patterns act as separating hyperplanes, which essentially multiply the rows of $X$ by 0 or 1, and can be represented by diagonal matrices. For fixed $X$, the set of all possible ReLU activation patterns may be expressed as

$$\mathcal{D}_X = \left\{ D = \text{diag}\left(\mathbf{1}(Xv \geq 0)\right) : v \in \mathbb{R}^d \right\}.$$

The cardinality of $\mathcal{D}_X$ grows as $|\mathcal{D}_X| = \mathcal{O}\left(r(n/r)^r\right)$, where $r := \text{rank}(X)$ Pilanci & Ergen (2020). Given $D_i \in \mathcal{D}_X$, the set of vectors $v$ for which $(Xv)_+ = D_i Xv$, is given by the following convex cone: $\mathcal{K}_i = \{v \in \mathbb{R}^d : (2D_i - I)Xv \geq 0\}$.

Unfortunately, $\mathcal{D}_X$ has exponential size Pilanci & Ergen (2020), which makes the convex reformulation based on a complete enumeration of $\mathcal{D}_X$ impractical. We can work with a subset of patterns based on sampling $P$ patterns from $\mathcal{D}_X$ to obtain a tractable convex reformulation. This leads to the following subsampled convex reformulation Mishkin et al. (2022); Bai et al. (2023):

$$\min_{(v_i, w_i)_{i=1}^P} \ell \left( \sum_{i=1}^P D_i X(v_i - w_i), y \right) + \beta \sum_{i=1}^P ||v_i||_2 + ||w_i||_2 \tag{3}$$
$$\text{s.t. } v_i, \, w_i \in \mathcal{K}_i \quad \forall i \in [P].$$

Although equation 3 is based on subsampling patterns in the convex reformulation, it can be shown under reasonable conditions that equation 3 still has the same optimal solution as equation 2 Mishkin et al. (2022). Moreover, the recent work of Kim & Pilanci (2024) shows that even when they do not agree, the difference is negligible. Therefore, we can confidently work with the tractable convex program in equation 3.

In this paper, we set $\ell$ to the mean-square error loss. The recent work Bai et al. (2018) has shown in this case that by adding slack variables, equation 3 can be written as:

$$\min_{v,s,u} ||Fu - y||_2^2 + \beta ||v||_{2,1} + \mathbb{I}_{\geq 0}(s) \quad \text{s.t. } u = v, \, Gu = s \tag{4}$$

Here, the matrix $F \in \mathbb{R}^{n \times 2dP}$ is block-wise constructed by $D_i X$ terms.

# 4 CONVEX DPO

In this section, we introduce the Convex DPO (CVX-DPO) algorithm.

## 4.1 OUR APPROACH

In standard DPO, the goal is to obtain a good policy that is aligned with human user preferences. To do this, DPO initializes the policy network $\pi_\theta$ with the weights of a pre-trained network. It then aligns the policy model by solving the optimization problem:

$$L_{\text{DPO}}(\pi_\theta; \pi_{\text{ref}}) = -\mathbb{E}_{(x, y_w, y_l) \sim \mathcal{D}} \left[ \log \sigma \left( \beta \log \frac{\pi_\theta(y_w|x)}{\pi_{\text{ref}}(y_w|x)} - \beta \log \frac{\pi_\theta(y_l|x)}{\pi_{\text{ref}}(y_l|x)} \right) \right]. \tag{5}$$

The optimization problem in equation 5 is a large-scale non-convex optimization, which can be challenging to solve.

To remove this challenge, we propose to replace $\pi_\theta$ with a convex two-layer model. Specifically, we take a pre-trained model $f_{\theta_{\text{pre}}}(x)$ and stack a two-layer convex neural network $g_{\Theta_1, \theta_2}^{\text{cvx}}$ on top to serve as a binary classifier. $g_{\Theta_1, \theta_2}^{\text{cvx}}$ is then trained by solving the convex optimization problem in equation 4. Letting $\theta = (\theta_{\text{pre}}, \Theta_1, \theta_2)$, the resulting policy is then given by:

$$\pi_\theta^{\text{cvx}}(y|x) := \frac{1}{1 + \exp \left( -y g_{\Theta_1, \theta_2}^{\text{cvx}} \left( f_{\theta_{\text{pre}}}(x) \right) \right)}.$$

However, rather than do preference optimization with the weights of the entire model $g_{\Theta_1, \theta_2}^{\text{cvx}} \circ f_{\theta_{\text{pre}}}$, we freeze the weights of $f_{\theta_{\text{pre}}}$ and freeze $\Theta_1$ in $g_{\Theta_1, \theta_2}^{\text{cvx}}$. We then finetune $\theta_2$, the weights of the last layer of $g_{\Theta_1, \theta_2}^{\text{cvx}}$, by solving the following modified DPO optimization problem:

$$\min_{\theta_2} L_{\text{CVX-DPO}}(\pi_{\theta_2}^{\text{cvx}}) := -\mathbb{E}_{(x, y_w, y_l) \sim \mathcal{D}} \left[ \log \sigma \left( \beta \log \frac{\pi_{\theta_2}^{\text{cvx}}(y_w|x)}{\pi_{\text{ref}}(y_w|x)} - \gamma \right) \right]. \tag{6}$$

What is the advantage of solving equation 6 over equation 5. The answer is computational tractability, as the following proposition shows equation 6 is convex.

**Proposition 1.** *The optimization problem in equation 6 may be written as:*

$$\min_{\theta_2} \mathbb{E}_{(x, y_w, y_l) \sim \mathcal{D}} \left[ \log \left( 1 + \exp \left( -\beta y_w \theta_2^T (\Theta_1 f_{\theta_{\text{pre}}}(x))_+ - \gamma \right) \right) \right]. \tag{7}$$

*Moreover, it is convex as equation 7 is a logistic regression problem in $\theta_2$.*

Theorem 1 shows that $L_{\text{CVX-DPO}}$ is convex. Thus, we can solve it to global optimality in polynomial time using efficient gradient-based optimizers.

We refer to procedure we have just described as the **C**onvex **D**irect **P**reference **O**ptimization (CVX-DPO) algorithm.

## 4.2 CONVEX DPO ALGORITHM

---

**Algorithm 1** Convex DPO (CVX-DPO)

---

**Require:** Dataset $(x, y_w, y_l)$, Pre-trained model $f_{\theta_{\text{pre}}}(x)$, penalty parameter $\rho > 0$
    Train $\pi_\theta^{\text{cvx}}$ to obtain $(\Theta_1, \theta_2)$ by solving equation 4 using ADMM($\rho$) (Algorithm 2).
    Freeze weights of the first layer $\Theta_1$
    Finetune weights of second layer $\theta_2$ by solving the convex minimization problem:

$$\min_{\theta_2} \ \mathbb{E}_{(x,y_w,y_l)\sim\mathcal{D}} \left[ \log \left( 1 + \exp \left( -\beta y_w \theta_2^T (\Theta_1 f_{\theta_{\text{pre}}(x)})_+ - \gamma \right) \right) \right]. \quad \triangleright \text{Solve with AdamW}$$

    **return** $(\Theta_1, \theta_2)$.

---

We formally present the pseudocode for CVX-DPO in algorithm 1. As described above, we first train a two-layer convex network on top of a pre-trained model. Once this is done, we obtain the policy model by solving a convex logistic regression problem using AdamW.

The key to making CVX-DPO efficient is using ADMM to train $\pi_\theta^{\text{cvx}}$, which we now discuss in detail.

### 4.2.1 EFFICIENTLY TRAINING THE CONVEX POLICY NETWORK VIA ADMM

---

**Algorithm 2** ADMM for Convex ReLU Networks

---

**Require:** penalty parameter $\rho$
    **repeat**
        $u^{k+1} \approx \operatorname{argmin}_u \left\{ \frac{1}{2}\|Fu - y\|^2 + \frac{\rho}{2}\|u - v^k + \lambda^k\|_2^2 + \frac{\rho}{2}\|Gu - s^k + \nu^k\|^2 \right\}$    $\triangleright$ Use CG
        $\begin{bmatrix} v^{k+1} \\ s^{k+1} \end{bmatrix} = \operatorname{argmin}_{v,s} \beta\|v\|_{2,1} + \mathbf{1}(s \geq 0) + \frac{\rho}{2}\|u^{k+1} - v + \lambda^k\|^2$    $\triangleright$ Primal update
        $\lambda^{k+1} \leftarrow \lambda^k + \frac{\gamma_\alpha}{\rho}(u^{k+1} - v^{k+1})$    $\triangleright$ Dual $\lambda$ update
        $\nu^{k+1} \leftarrow \nu^k + \frac{\gamma_\alpha}{\rho}(Gu^{k+1} - s^{k+1})$    $\triangleright$ Dual $\nu$ update
    **until** convergence

---

To train $\pi_\theta^{\text{cvx}}$, we solve the optimization problem in equation 4 with the Alternating Direction Method of Multipliers (ADMM) Boyd et al. (2011). Our choice of ADMM for solving equation 4 is predicated on three important properties: 1) It possesses a robust convergence guarantee, 2) it is insensitive to hyperparameter settings, and 3) it is highly amenable to hardware acceleration. For the moment, we focus on 2) and 3). We return to 1) in Section 4.3.

Algorithm 2 formally presents ADMM for solving eq. (4). The main work consists of the first two lines, where two subproblems must be solved. The remaining lines only require vector addition and one matrix-vector product. The structure of these subproblems is what makes ADMM so compatible with hardware accelerators. The $u^{k+1}$-subproblem is simply a regularized least-squares problem. This can be readily solved with the Conjugate Gradient (CG) algorithm, which only requires highly parallelizable matrix-vector products (matvecs) with the dense matrices $F$ and $G$. GPUs excel at accelerating matvecs and other linear algebraic primitives. This is further amplified by using JAX's JIT compilation feature, which enables us to compile CG for solving the $u^{k+1}$-subproblem, leading to very fast solve times. Moreover, this problem can be solved inexactly and ADMM will still converge. Thus, the linear solve is not a bottleneck. The $(v^{k+1}, s^{k+1})$ subproblem corresponds to the proximal operator for the group Lasso and has an analytic solution that may be computed in $\mathcal{O}(dP)$ time.

The other advantage of ADMM is that it only has one hyperparameter $\rho > 0$, to which its convergence is relatively insensitive. Indeed, in contrast to gradient methods, we shall see in theorem 2 that ADMM converges for any $\rho > 0$. Thus, $\rho$ does not require aggressive tuning in order for ADMM to yield good performance. Moreover, $\rho > 0$ can automatically be tuned using a technique known as *residual balancing* (Boyd et al., 2011). This is a common technique for setting $\rho$ in existing ADMM solvers (Stellato et al., 2020; Schubiger et al., 2020; Diamandis et al., 2023).

### 4.3 Convex-DPO: Convergence Guarantees

As CVX-DPO solves convex optimization problems, it immediately inherits the rich convergence theory associated with convex optimization algorithms. We begin with the following result: ADMM converges ergodically at an $\mathcal{O}(1/k)$-rate.

**Theorem 2** (Convergence of ADMM for equation 4). *Let $\{\delta_k\}_{k \geq 1}$ be some summable sequence. Run Algorithm 2 and suppose at each iteration the computed $u^{k+1}$ satisfies:*

$$\left\| u^{k+1} - \mathrm{argmin}_u \left\{ \frac{1}{2} \|Fu - y\|^2 + \frac{\rho}{2} \|u - v^k + \lambda^k\|_2^2 + \frac{\rho}{2} \|Gu - s^k + \nu^k\|^2 \right\} \right\| \leq \delta_k.$$

*Then after $K$ iterations, the output of ADMM algorithm 2 satisfies:*

$$\|F\bar{u}^K - y\|^2 + \beta \|\bar{v}^K\|_{2,1} + \mathbf{1}(\bar{s}^K \geq 0) - p^\star = \mathcal{O}(1/K),$$

$$\left\| \begin{bmatrix} I_{2dP} \\ G \end{bmatrix} \bar{u}^K - \begin{bmatrix} \bar{v}^K \\ \bar{s}^K \end{bmatrix} \right\| = \mathcal{O}(1/K).$$

The proof follows from general convergence results for ADMM and is provided in the supplement. Theorem 2 shows that ADMM converges ergodically to the global minimum of equation 4 at an $\mathcal{O}(1/k)$-rate. Moreover, this is guaranteed for any $\rho > 0$ and when the $u$-subproblem is solved inexactly. Consequently, ADMM's convergence is very robust. This stand

The Convex-DPO minimization objective in equation 7 is smooth, convex, and has a Lipschitz continuous gradient. The latter property follows as the logistic loss has a Lipschitz continuous gradient. Thus, we can apply Accelerated Gradient Descent (AGD) (Nesterov, 1983; d'Aspremont et al., 2021), which has the worst-case optimal convergence rate, to solve equation 7.

**Theorem 3** (Efficient minimization of CVX-DPO loss eq. (7)). *Suppose we run AGD to solve equation 7. Then after $k$ iterations, the output $\theta_2^k$ satisfies:*

$$L_{\mathrm{CVX\text{-}DPO}}(\pi_{\theta_2^k}^{\mathrm{cvx}}) - \min_{\theta_2} \; L_{\mathrm{CVX\text{-}DPO}}(\pi_{\theta_2}^{\mathrm{cvx}}) = \mathcal{O}(1/k^2).$$

Theorem 3 shows we can train the CVX-DPO loss to global optimality in polynomial time. This contrasts greatly with DPO, which is non-convex and lacks convergence guarantees. Moreover, tuning the optimizer in DPO can be difficult, resulting in poor performance (Meng et al., 2024). As full-gradient methods like AGD are expensive for large-scale training, we used AdamW (Loshchilov & Hutter, 2017) in this paper. Unfortunately, AdamW does not inherit this guarantee. But we have found it works well in our setting without tuning. In future work, we would like to explore variance-reduced stochastic gradients like in Frangella et al. (2024), which can be as computationally efficient as Adam, but come with much stronger convergence guarantees.

## 5 Experiments

We experiment with 4 models (DistilGPT, GPT2, GPT2-M, FLAX-GPT2) on 3 datasets. Our goal is to examine the effectiveness of DPO to train a small language model on one GPU, and to see if we can make the process even more cost effective by providing more signal with the ADMM optimized convex neural network.

### 5.1 Datasets

This study explores three datasets: both synthetically generated and well-established datasets to be consistent with previous work. Each dataset is selected to offer a different qualitative assessment of

the methodology. We format each dataset into "prompt", "chosen", "rejected" labels to be consistent with the original DPO paper. Appendix B contains examples of the training dataset, as well as generated samples. In each case we follow the DPO dataset of preferences format with $\mathcal{D}$ be defined as follows: $\mathcal{D} = \{x^{(i)}, y_w^{(i)}, y_l^{(i)}\}_i^N$. Where $y_w^{(i)}$ is the "chosen" output and $y_l^{(i)}$ is the "rejected" output. The key difference in our custom preference data generation strategy is that we utilize only the natural conversational data between two agents. Therefore we select the first utterance as "prompt", then the following response as "chosen" with the next following responses as "rejected". This has the following advantages: eliminates the need to prompt an external LLM to generate the rigid chosen-rejected dataset format expected by DPO, naturally keeps the dataset in the same distribution since all samples occur within the same conversation, allows greater ease of using vast conversational datasets without complex processing and selection.

- **IMDb Sentiment Generation** This dataset contains a collection of positive and negative movie reviews from IMDb (Tripathi et al. (2020)) for the task of controlled sentiment generation. This is selected as the baseline dataset for all methods, to be consistent with the original DPO paper and verify mode implementations. In this case $x$ is the title of a movie, and $y$ is the generated positive sentiment, which should also accurately reflect the movie.

- **Educational Tutor Dataset** This is a custom generated task-orientated dataset in an educational setting. Please see Appendix B for data samples. In each conversation we create 4000 dialogue prompts with GPT-3.5 (Achiam et al. (2023)) then use 2 instances of agents to simulate conversations a student studying for a quiz and a tutor assisting. The dataset is formatted as Prompt, Agent 1, Agent 2, Agent 1, etc. We then create the DPO dataset with $y_w^{(i)}$ as the completion immediately following the agent query, and $y_l^{(i)}$ of the alternative agent's generation 2 steps forwards from the guests query. The creation of this dataset serves 2 purposes: Since real world applications often provide limited or unlabeled data, we are interested in how well human preferences can be optimized with a simulated real world dataset in a well-defined hospitality setting. Secondly, since this is the smallest of our three datasets, we are interested in the possibility of aligning LMs to human preference with very little data as described by the authors of Zhou et al. (2024). We prompt the student agent to ask questions across the following areas of study: math, science, history, literature, art, geography, biology, physics, chemistry, music, mythology, astrology, literature, philosophy, and chess.

- **Stanford-SHP** This is the largest dataset in our experiments, and is selected to stress test the memory and speed performance of our models on the setup described in section 5.2. The Stanford-SHP (Ethayarajh et al. (2022)) is a dataset of 385K collective human preferences over responses to questions in 18 different subject areas. This dataset also serves to generate preferable responses to prompts, however due to the slow iteration and sample during eval limits, we are more interested in how it affects our systems compute and qualitative generative output performance.

## 5.2 EXPERIMENTAL DETAILS

Throughout all experiments we use DistilGPT2(Li et al., 2021), GPT2 (Radford et al., 2019), GPT2-Medium architecture, and FLAX-GPT2 as the policy model. CVX-DPO does not require a reference model for stability, and instead use the convex neural network (NN) to signal the model parameters towards "chosen" log probabilities. Our selection of GPT based policy models is due to its versatility to run in both JAX and Pytorch frameworks, while utilizing a small number of 82 million parameters in the form of DistilGPT2. This architecture in particular retains approximately $97\%$ of GPT-2's language understanding skills despite its reduced size. All experiments are run singularly on Ubuntu 22.04 with one RTX-4090, CUDA 12.6 and Jax 0.4.33. Maximum training time reached 2.15 hours on the Stanford-SHP with the DPO loss, while minimum training time occurred with the custom Educational-Tutor dataset in supervised fine-tuning mode of approximately 2min. We keep the same learning rate and configurations as the official DPO implementation.

## 5.3 DPO WITH CONVEX-NN FEEDBACK

In this section we describe the model implementations and benchmarked methods of this work. For each model, we train and evaluate on the Educational Tutor Dataset, then the IMDb Sentiment

dataset as described in section 5.1. The Stanford-SHP dataset is only trained and evaluated on the JAX-DPO re-implementation of the official source code by (Rafailov et al., 2024). All other source code, including all JAX code, is custom coded by the authors of this project. During evaluation, metrics and training loss are monitored on Weights and Biases (Jocher et al. (2021)), validated with custom implemented functions for measuring TFLOPs and VRAM, then during human evaluation we sample from our frozen and custom trained models.

**Baseline** The baseline model is DistilGPT2 with supervised fine-tuning loss. In subsequent DPO settings, we initiate from the saved model of the SFT baseline for each model.

**DPO Method** Next we train and evaluate on the traditional DPO model. The reference model is essentially frozen, and we optimize the policy model with the DPO loss. Rafailov et al. (2024) provide in-depth analysis on the mechanics of DPO. This step is significantly more memory and time intensive than the baseline model. Naive implementations of the DPO model in Pytorch are not able to complete training on our dataset due to compute limitations of the experiment setting in section 5.2. Therefore we re-implement this train loop in JAX, remove Fully-sharded-data-parallel (FSDP), and rewrite utilities to load our custom features. The addition of the reference model to stabilize training incurs both memory and compute costs which are significant.

**SimPO Method** This is a recent reference-free method that has shown to outperform DPO on several metrics. However the introduction of the additional hyperparameter is crucial to its success, and thus results in longer training time.

**CVX-DPO Algorithm** Our novel algorithm builds on prior work, where we have seen that the combination of the convex ReLU NN implemented with ADMM in JAX is able to handle datasizes such as ImageNet (Recht et al. (2019)) and IMDb and yields faster convergence with solutions of better quality. We are motivated by the DPO objective, which treats the *policy optimization task as a binary classification* with cross-entropy problem. Therefore, what if we can speed up the optimization of the DPO loss by giving it auxiliary signal with the convex-NN model?

In observing the DPO loss, we note that the main component is the inner log ratio between the policy model and the reference model, and then difference in log ratios between "chosen" and "rejected". We conjecture that by extracting the hidden features as the policy model optimizes, we should be able to leverage the convex-admm method to solve the binary classification problem. The output of the convex model provides optimal weights and classification metrics, therefore we label all "chosen" = 1 and all "rejected" = 0 in training to optimize for user preferences. This convex block is then added into the training loop of the DPO training pipeline, and used to optimize DPO's BCE style loss by giving strong reward signal feedback.

The official implementation of DPO uses RMSProp (Shi & Li (2021)), which is seen to be as performant as Adam Kingma & Ba (2014) but more memory efficient since it requires less storage variables. However we note that the integration of the convex-DPO algorithm can provide advantages such as robustness against hyperparameter tuning and faster convergence. This aims to push the DPO loss towards a more globally optimal solution even faster. Please see Appendix C for performance plots.

## 5.4 EVALUATION BENCHMARKS

his is because human preference is hard to define, and recent work of Celikyilmaz et al. (2020) has shown that often humans will prefer simply the longer generated output without reason. Therefore we qualitatively evaluate our output with the 25 human participants. This is also in order to be consistent with existing literature of the seminal DPO paper. We structure evaluation as follows:

- vary temperature hyperparameter ($T$) from 0.1 to 0.4.
- for each temperature, in each of the 3 models listed in section 5.3 above, we input the same 12 prompts. For example, a prompt might be "What is the structure of a Shakespearean sonnet?"
- each of the models generate a response, which is shuffled into a multiple choice survey, and sent to 25 human volunteers
- We record each human's preferences in selection, and vary the sequence of model generated output in multiple choice questions in order to mitigate human bias.

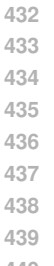

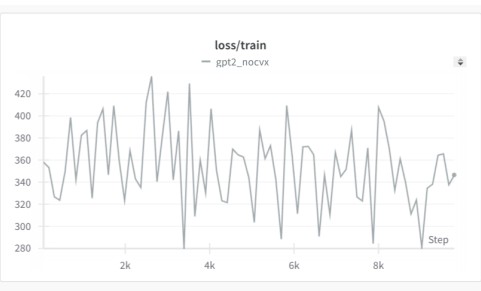

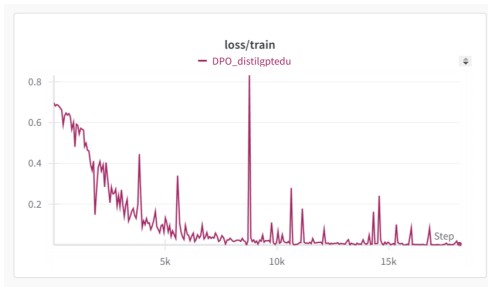

Figure 1: DPO Validation Loss without Ref Model

Figure 2: DPO-Convex Training Loss

Table 1: Feedback from 25 Human Volunteers

|  | FST | DPO | CVX-DPO |
|---|---|---|---|
| IMDb (Win rate) | 1 | 3 | 4 |
| Education (Win rate) | 0 | 3 | 3 |
| IMDb Avg Win % | 72.2% | 62.5% | 68.5 |
| Education Avg Win % | 0 | 68.7% | 83.3% |

Table 5.5 summarizes the average performance of each model. The Educational Tutor response survey section asked users to select the response that was the most HELPFUL and HUMAN, as if the participants were studying for a quiz with a tutor. This is the most meaningful response, since it utilizes our alternating preference dataset generating technique.

We also measure the speed and stability of training, as well as the robustness to hyperparameter tuning on the convex setting.

Finally we observe training time, loss achieved (see below), and difference in scalability between frameworks.

## 5.5 RESULTS

In this section we compare the results of the 3 models discussed in section 5.3. Although we perform ablation studies with varying $0.1 < T < 0.4$. The baseline model is consistently the fastest to train, although it consistently demonstrates the highest amount of repetition in its output. This is further validated in our human feedback survey, where the baseline model won on only one out of thirteen questions.

The DPO-Convex model shows the most stable training performance. Despite variances in hyper-parameters such as temperature, data size, batch size, this model was consistently able to stably and quickly decrease in loss. Figures 1 and Figure 2 show the training performance of the DPO-Convex model without any tuning of hyperparameters.

Table 5.5 summarizes the results of the human feedback survey, and shows both win rate and the average preference of each model. The average preference is calculated as the percentage of each model's win rate divided by the number of times it won. We provide the average preference percentage as a metric since it gives better signal as to how preferred a model was. For example, the baseline FST model only won on one question, but was strongly preferred in that case by most humans. The Educational Tutor dataset saw an equal win-rate count between the DPO model and DPO-Convex model, but humans had stronger preference to the answers of DPO-Convex (83.3%).

## 6 ANALYSIS AND DISCUSSION

Since we use smaller datasets on the DistilGPT2 Li et al. (2021) model, we expected to see a certain amount of repetition in the output. This is most prominent in the baseline FST model. For example, the prompt "How can I check in? The answer is yes. I can't....". Although we vary $T$ and its effect

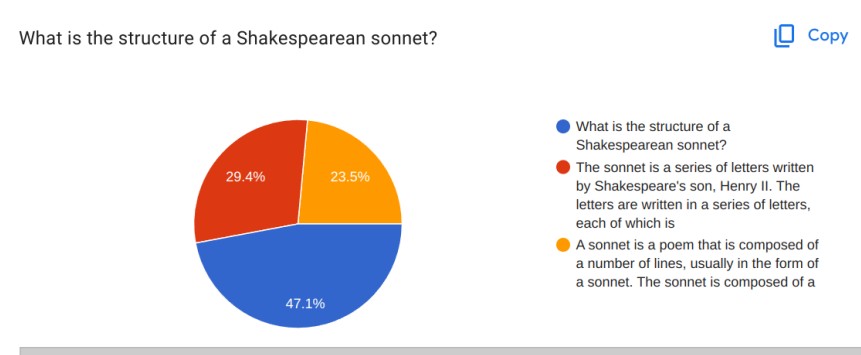

Figure 3: DPO with convex wins with T=0.1

on perplexity from $0.001$ to $2.01$ in steps of $0.3$, the baseline FST does not increase in performance and is consistent in its repetition (as seen in C).

The DPO model needed approximately double the amount of time to train as the baseline FST model. However the DPO model notably generated varying degrees of creativity in the same prompt as temperature varied. We note that the DPO model instances that won on the human feedback survey were all instances where $T < 0.4$. This is in contrast to our conjecture that higher temperature will produce more desirable results with DPO since humans prefer more creative output. We also note that since our generated dataset is in a Hotel-Concierge setting, it's possible humans prefer more consistency versus creativity. In the two sample questions posed to our human volunteers, it is clearly seen that the baseline model shows repetition, but the DPO model is preferred with $T = 0.601$. The DPO-Convex model tends to generate longer responses. However this might be attributed to its capacity for faster training.

The DPO-Convex model showed the most stable training performance. While training on one GPU and without compromising dataset size, loss was able to consistently go down regardless of varying hyperparameters. This agrees with our conjecture that adding the convex feedback increases robustness, and eliminated the need to continue with further hyperparameter tuning in experiments with the DPO-convex model. Please see Appendix C for training plots. In human feedback, both DPO and the DPO-Convex model were almost equally preferred. We attribute this to small sample size of questions and volunteers, and realize the significance and difficulty of evaluating preference generation. This direction leaves room for more future work.

## 7 CONCLUSION

In this work we introduce CVX-DPO, a novel algorithm for preference learning using convex optimization. This significantly improves the robustness and speed of convergence of the convex auxiliary signal with the DPO objective. The resulting algorithm is more robust to hyperparameter tuning (such as learning rate), and allows fast iteration with preferable output on minimal VRAM consumption. The ADMM solve method provides further speed, efficiency, and parallelism. We implement our methods in JAX and run experiments run on one GPU for speed and better memory efficiency. Our custom generated synthetic dataset of the Educational Tutor setting simulates real world conversational data with turns of phrase, as opposed to many preference learning datasets which sample acute "chosen" and "rejected" responses to a single prompt. We validate the efficacy of our fast lightweight pipeline against 25 human volunteers, with promising results. Thus we hope this work can reduce the barrier for entry even more for individual researchers and for various educational purposes, and take a step towards democratizing the large language model regime.

**Limitations and Future Work** Future work will involve running our JAX experiments on TPUs or GPU clusters. Since JAX and ADMM were developed with easy parallelization in mind, more performant scaling results should be explored where we can handle even more data.

## 8 REPRODUCIBILITY STATEMENT

Our JAX code base is available for reproducibility, with configurations to replicate experiments. We provide both the original custom generated 4000 conversations in the Educational Tutor dataset, as well as the preference alignment version with the alternating strategy. All other datasets utilized are publicly available, and we adhere to the original DPO hyperparameters to ensure consistency. It is suggested to replicate our experiments on NVIDIA GPUs with Ubuntu version 22.04, JAX version 0.4.33, CUDA 12.6, NVIDIA Drivers 560 and above.

## 9 ETHICS STATEMENT

Our main objective in this work is to make preference alignment in language models more easily accessible to individual researchers, students, and the more general populace. We strongly believe that taking a small step towards democratizing research in language model capabilities is also a meaningful step towards and ethical AI future. We hope this work can assist more individuals to be both interested in LMs, and also support more educational purposes.

### AUTHOR CONTRIBUTIONS

All authors contributed equally on this work.

### ACKNOWLEDGMENTS

To be omitted for the time being in acknowledgement of anonymity during review.

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

# A    PROOFS OF MAIN RESULTS

## A.1    PROOF THAT CONVEX DPO LOSS IS CONVEX.

Recall the CVX-DPO objective is given by:

$$\min_{\theta_2} - \mathbb{E}_{(x,y_w,y_l)\sim\mathcal{D}} \left[ \log \sigma \left( \beta \log \frac{\pi_{\theta_2}^{\text{cvx}}(y_w|x)}{\pi_{\text{ref}}(y_w|x)} - \beta \log \frac{\pi_{\theta_2}^{\text{cvx}}(y_l|x)}{\pi_{\text{ref}}(y_l|x)} \right) \right].$$

For CVX-DPO, we have that

$$\pi_{\theta_2}^{\text{cvx}}(y_w|x) = \frac{1}{1 + \exp\left(-y_w(\theta_2^T \tilde{x})\right)},$$

where $\tilde{x} = (\Theta_1 x)_+$. Using this expression for $\pi_{\theta_2}^{\text{cvx}}(y_w|x)$, we find

$$\beta \log \frac{\pi_{\theta_2}^{\text{cvx}}(y_w|x)}{\pi_{\text{ref}}(y_w|x)} - \beta \log \frac{\pi_{\theta_2}^{\text{cvx}}(y_l|x)}{\pi_{\text{ref}}(y_l|x)} = \beta \log \left( \frac{\pi_{\text{ref}}(y_l|x)}{\pi_{\text{ref}}(y_w|x)} \frac{\pi_{\theta_2}^{\text{cvx}}(y_w|x)}{\pi_{\theta_2}^{\text{cvx}}(y_l|x)} \right)$$

$$= \beta \log \left( \frac{\pi_{\text{ref}}(y_l|x)}{\pi_{\text{ref}}(y_w|x)} \frac{\pi_{\theta_2}^{\text{cvx}}(y_w|x)}{1 - \pi_{\theta_2}^{\text{cvx}}(y_w|x)} \right)$$

$$= \beta \log \left( \rho_{\text{ref}}(x) \exp\left(y_w \theta_2^T \tilde{x}\right) \right)$$

$$= \beta \log(\rho_{\text{ref}}(x)) + \beta y_w \left(\theta_2^T \tilde{x}\right)$$

From this last display and the definition of the sigmoid function, it immediately follows that:

$$\log \sigma \left( \beta \log \frac{\pi_{\theta_2}^{\text{cvx}}(y_w|x)}{\pi_{\text{ref}}(y_w|x)} - \beta \log \frac{\pi_{\theta_2}^{\text{cvx}}(y_l|x)}{\pi_{\text{ref}}(y_l|x)} \right) = -\log(1 + \exp\left(-\beta y_w \left(\theta_2^T \tilde{x}\right) - \beta \log(\rho_{\text{ref}}(x))\right).$$

Thus, using the last display and the definitions of $\tilde{x}$ and $\rho_{\text{ref}}(x)$, the Convex DPO objective may be rewritten as:

$$\min_{\theta_2} \mathbb{E}_{(x,y_w,y_l)\sim\mathcal{D}} \left[ \log \left( 1 + \exp \left( -\beta y_w \theta_2^T (\Theta_1 x)_+ + \beta \log \frac{\pi_{\text{ref}}(y_w|x)}{\pi_{\text{ref}}(y_l|x)} \right) \right) \right].$$

We see that the Convex DPO objective is a logistic regression problem in $\theta_2$, and thus is convex.

# B    EXAMPLES OF DATA AND TRAINING

Figure 4: Example of custom generated Educational Tutor conversation dataset

# C    PERFORMANCE PLOTS

Please see the following images for performance plots.

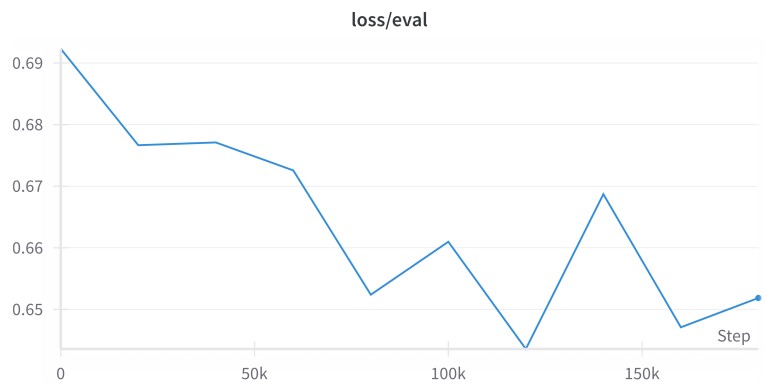

Figure 5: Training for DPO naive

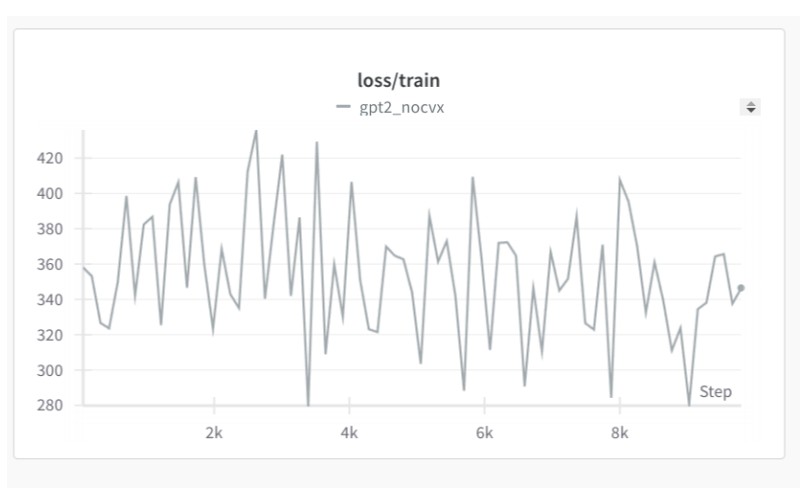

Figure 6: Larger image of training for DPO as discussed above

```
temperature      prompt                           generated_text
tensor(0.0010) How can I check in?               How can I check in?The answer is yes. I can't. I can't. I can't. I can't. I can't. I can't.
tensor(0.0010) Is room service available?        Is room service available?The answer is yes. The service is available for all ages. The ser
tensor(0.0010) What is the weather like today?   What is the weather like today?The weather is so cold that it's hard to see the sun. The we
tensor(0.0010) Is there a swimming pool?         Is there a swimming pool?"
tensor(0.0010) Do you have valet parking?        Do you have valet parking?The answer is yes. It's a good idea to have a parking lot in your
tensor(0.0010) Do you have a gym?                Do you have a gym?The answer is yes. It's a gym that's not a gym. It's a gym that's not a g
tensor(0.0010) Are you pet friendly?             Are you pet friendly?The answer is yes.
tensor(0.0010) Where is the nearest park?        Where is the nearest park?The park is a little bit of a mystery, but it's worth a look. The
tensor(0.0010) Can I request late checkout?      Can I request late checkout?The answer is yes. I have no idea. I have no idea what the pric
tensor(0.0010) Do you have airport transfers?    Do you have airport transfers?The airport is a great place to go. It is a great place to go
tensor(0.3010) How can I check in?               How can I check in?The answer is yes. I can't. I can't. I can't. I can't. I can't. I can't.
tensor(0.3010) Is room service available?        Is room service available?The answer is yes. The service is available for all ages. The ser
tensor(0.3010) What is the weather like today?   What is the weather like today?The weather is so cold that it's hard to see the sun. The we
tensor(0.3010) Is there a swimming pool?         Is there a swimming pool?"
tensor(0.3010) Do you have valet parking?        Do you have valet parking?The answer is yes. It's a good idea to have a parking lot in your
tensor(0.3010) Do you have a gym?                Do you have a gym?The answer is yes. It's a gym that's not a gym. It's a gym that's not a g
tensor(0.3010) Are you pet friendly?             Are you pet friendly?The answer is yes.
tensor(0.3010) Where is the nearest park?        Where is the nearest park?The park is a little bit of a mystery, but it's worth a look. The
tensor(0.3010) Can I request late checkout?      Can I request late checkout?The answer is yes. I have no idea. I have no idea what the pric
tensor(0.3010) Do you have airport transfers?    Do you have airport transfers?The airport is a great place to go. It is a great place to go
tensor(0.6010) How can I check in?               How can I check in?The answer is yes. I can't. I can't. I can't. I can't. I can't. I can't.
tensor(0.6010) Is room service available?        Is room service available?The answer is yes. The service is available for all ages. The ser
tensor(0.6010) What is the weather like today?   What is the weather like today?The weather is so cold that it's hard to see the sun. The we
tensor(0.6010) Is there a swimming pool?         Is there a swimming pool?"
tensor(0.6010) Do you have valet parking?        Do you have valet parking?The answer is yes. It's a good idea to have a parking lot in your
tensor(0.6010) Do you have a gym?                Do you have a gym?The answer is yes. It's a gym that's not a gym. It's a gym that's not a g
tensor(0.6010) Are you pet friendly?             Are you pet friendly?The answer is yes.
tensor(0.6010) Where is the nearest park?        Where is the nearest park?The park is a little bit of a mystery, but it's worth a look. The
tensor(0.6010) Can I request late checkout?      Can I request late checkout?The answer is yes. I have no idea. I have no idea what the pric
tensor(0.6010) Do you have airport transfers?    Do you have airport transfers?The airport is a great place to go. It is a great place to go
tensor(0.9010) How can I check in?               How can I check in?The answer is yes. I can't. I can't. I can't. I can't. I can't.
```

Figure 9: FST baseline resistant to varying T, consistently repeats output

## D    HUMAN EVALUATION

We have 25 human volunteers selecting their most preferred generated output. The survey is conducted as a total of 13 questions, across 2 datasets (5.1), with output generated by each of the three models discussed in 5.3. Further details of survey human evaluation is summarized in 5.5. The raters

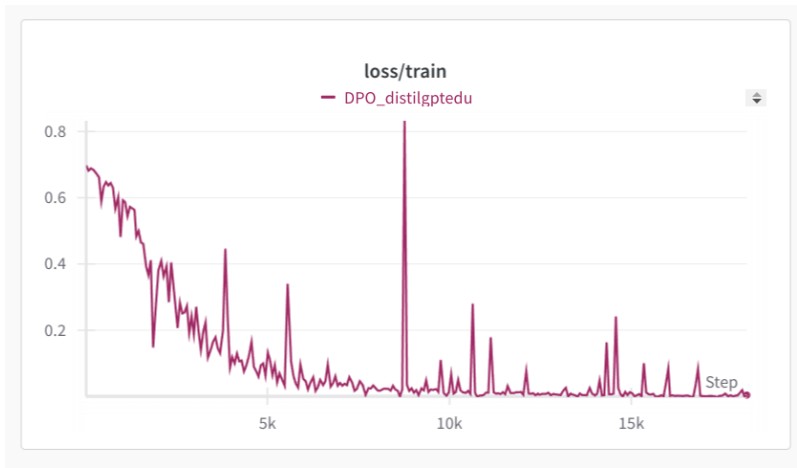

Figure 7: Sample of training for DPO-Convex

were Stanford students (from graduate to Ph.D.), University of Toronto students (Ph.D.), Google Software Engineers, and medical practitioners in Veterinary science. We gratefully acknowledge the contribution of each of our human subjects, listed in random order: Anna Goldie, Kevin Nam, Zhong Wei Dang, Shaun Benjamin, Sera Benjamin, Yue Benjamin, Noriyuki Shintoku, Jenny Song, Zachary Frangella, Stephen Sapperton, Mary Habib, Scarlet Arreola Barrones, Farhis Kordi, Adam Wong, Jia Yu, Tong Zhang, Juan Jaramillo, and Trevor Germain.

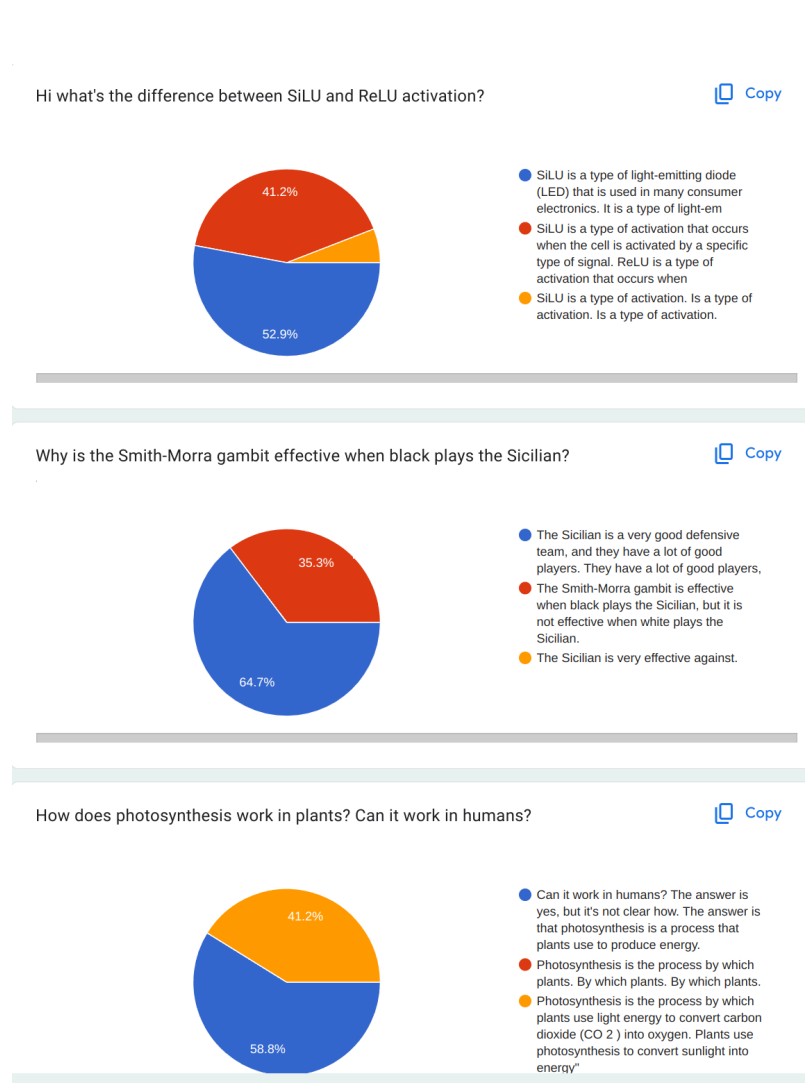

Figure 8: Sample of Survey to Human Subjects

