# OpenReview forum: "Scalable Preference Learning for Large Language Models via Convex Optimization"
_ICLR.cc/2025/Conference — ICLR 2025 Conference Withdrawn Submission_

### Official Review · Reviewer_GGkp · 2024-10-31

**Soundness:** 2
**Presentation:** 1
**Contribution:** 2
**Rating:** 3
**Confidence:** 3

**Summary:**

The paper presents a method using a convex reformulation of NNs to perform lightweight DPO and introduces a strategy for generating preference datasets. They use a convex NN that classifies hidden features as chosen or rejected and uses the output with a DPO style loss. The dataset generation strategy uses a natural conversation between two models and sets the first part of the dialogue as the prompt, the second as the chosen responses, and the remaining parts as rejected.

**Strengths:**

The paper introduces a reference-free method for direct preference learning, and uses a convex-NN for more efficient training of a two layer NN block.

**Weaknesses:**

There are parts of the notation that are unclear such as in line 145, $X$ is said to be a two-layer ReLU MLP but has a matrix shape. Additionally, in equation (4a), F is defined in terms of $D_i X$ where $D_i$ had not been defined.

The method itself is also not clearly explained. They mention extracting hidden features as the policy model optimizes, but it is not described how/where these are extracted from. They also mention optimizing DPO's BCE style loss but it is not clear exactly how they define the loss given that the method should be reference-free. Furthermore, it is unclear why given the same reference-free loss, why the reward should be defined using a convex-NN instead of the original log-prob definition. Along with this, it is unclear how this method performs with respect to existing reference-free methods such as ORPO or SimPO.

Lastly, it is unclear as to the advantage of the dataset generation method. One strong issue is using most of the conversation as rejected responses which contrasts a direct response to the prompt with responses that would not appear in the same context. Additionally, the novelty of the method is unclear as datasets such as HH already draw multiple samples from the same conversation.

**Questions:**

1. Can you define each of the terms in equation 4 more clearly along with $D_i$ and $X$?
2. How does this method compare to existing reference-free methods?
3. What is the motivation for using the convex-NN to define reward?

---

> ### Author Response · Authors · 2024-12-04
>
> Thank you for your feedback! Details about the notation has been fixed, and variables such as D and X are now defined in the convex reformulation Section 3 to give background.
>
> The method is now more clearly explained. We re-wrote and reorganized the paper to include a clear section naming and explaining the CVX-DPO novel algorithm. There is now a background section explaining convex neural networks, convergence proofs, and the algorithm box. The objective is listed and compared against other objectives in the table above, and we have included the derivation and formulation of our objective function. We have added SimPO as a competing method now in our experiments. The motivation for the convex NN is that it gives polynomial time convergence guarantees and that it is more resource and time efficient to solve. Please see metrics regarding this above.
>
> The dataset generation method is intended to simulate a real world dataset conversational situation. It offers diverse topics, and is varying turns of phrase that may occur in an educational setting. The preference generative sampling procedure is intended to relieve some of the dependence on formulating DPO training datasets into a chat template style, which is often sensitive and requires extensive training. We have added experiments against SimPO, and agree with the authors of SimPO that existing ref-free methods still require crucial hyperparmeter tuning to succeed [1]. In contrast, CVX-DPO does not exhibit this weakness, and is faster while requiring less TFLOPs for increased efficiency. Our method also provides convergence guarantees in polynomial time, now listed in Section 3.

---

### Official Review · Reviewer_yuu8 · 2024-11-02

**Soundness:** 1
**Presentation:** 1
**Contribution:** 1
**Rating:** 1
**Confidence:** 4

**Summary:**

This paper introduces a combination of
- convex formulation of MLP
- alternating direction method of multipliers
as an alternative to gradient descent to solve the DPO alignment.
The authors claim that with the ADMM formulation, the parallelization efficiency can be increased. They implement the method in JAX and is able to train on a single RTX-4090 GPU.

**Strengths:**

The presentation of this paper is difficult to understand, as such, I don't have enough understanding to comment on the strengths.

**Weaknesses:**

There are many obvious reasons to recommend a rejection to this paper, here I list a few.
- This major contribution claimed by this work is the convex optimization & ADMM algorithm to perform DPO. But there is no information provided on which part of the DPO requires a MLP that is convexified, and which loss is solved in ADMM instead of SGD. The concepts of convex-nn (1)(2), admm (4) and DPO (5) are introduced as disconnected pieces, there is no explanation on what each symbol means, e.g. F & G in (4), what do they represent in the context of DPO etc. Unfortunately I could not understand what this method is doing given the current presentation.
- I find it difficult to understand the convex-nn part without referring to other papers, I would suggest reduce the text introducing JAX to save some space for a preliminary introduction of convex-nn, as it seems more relevant to the main method.
- The experimental report only human evaluations, which is not the standard in relevant literature, which often use LLMs as evaluators. Human evaluations could have strong subjection variance especially in a small number (17).
- While this work is motivated from the robustness of training, the speed, and the capability to run on a single GPU. The experiments report win rate over DPO. I'm not convinced why this method would out-perform DPO if the main difference is optimization algorithm.

**Questions:**

IMO this paper needs a major revision in the writing. I can not ask meaningful question with the current level of understanding.

---

> ### Author Response · Authors · 2024-12-04
>
> While we politely disagree with your point, we have addressed them in the general posted comments above, thank you for your feedback!

---

### Official Review · Reviewer_DuL5 · 2024-11-03

**Soundness:** 2
**Presentation:** 1
**Contribution:** 2
**Rating:** 3
**Confidence:** 3

**Summary:**

This paper proposes a lightweight alternative to RLHF in large language models with convex optimization. It reformulates the DPO objective using convex neural networks and leverages ADMM, achieving efficient training on a single GPU. Also, the authors implement the method in JAX, which improves the memory efficiency. Empirically, three datasets were explored by comparing DPO and DPO-convex.

**Strengths:**

DPO-convex is a less computationally intense alternative compared to DPO, thus lowering the hardware requirements for preference learning and making RLHF more accessible.

**Weaknesses:**

1.	The writing of this paper is not clear and the presentation can be significantly improved. In particular, it is not clear what the objective is for DPO-convex. How is a convex-nn constructed to formulate this objective? Please state the method and the setting in a more formal tone, and provide a detailed introduction on the methods or techniques used. For instance, how the convex-NN is constructed and integrated with the DPO objective?

2.	The authors mention implementing their method in JAX for improved memory efficiency, but no code is provided. Also, there are no quantitative comparisons on the memory usage of these two methods. Please provide a link for the code repository and include specific memory usage metrics for both methods in the results section.

3.	The models fine-tuned in the experiments are relatively small (DistilGPT2, GPT-2, GPT-2-medium). The scalability of this method to SOTA large models is uncertain. It would be great if the authors can conduct experiments on these models like Mistral or Llama 3 and more complicated tasks like MT bench or Alpaca-Eval?

4.	A minor issue is the inconsistency in the reported number of volunteers, which is stated as 17 in some instances and 25 in others. Please see lines 053 and 069.

**Questions:**

1.	What does “prox” mean in equation (4b)?

2.	Could you explain what “FST” stands for in the paper?

3.	The results in Table 1 are somewhat unclear. Could you elaborate on what a win rate of 1, 3, or 4 represents?

---

> ### Author Response · Authors · 2024-12-04
>
> Thank your feedback! We have thoroughly re-written to paper to address your questions and concerns. The initial typo of 17 persons was due to some discussion about the opinions of our human volunteers to being named in the work. We have resolved this problem in the revision.
>
> * With regards to point 1, we have added Section 3 to give background, mathematical definitions, and clearly derive our objective. This also resolves points 1 and 2 of your questions. We have also added convergence guarantees for our novel algorithm to  global optimality in polynomial time.
> * Our JAX codebase was provided in the zipped upload during submission. We have now also added an anonymous github repo for your convenience. Please see the link above in general comments.
> * Our goal is to push the boundaries of what can be achieved on one GPU at this time. The inception of this project arose from the realization that DPO and its variants ran into OOM issues on one RTX-4090, on very small models and datasets. Additionally extensive formatting of the preference dataset into chat template form is required, yet even after extension training results are often unstable. This is discussed in greater detail in the recent work of [1] and [2]. By optimizing on one GPU in JAX, we are able to lift the memory and resource constraints with the goal to democratize preference learning. The advantage of our JAX codebase is that it allows highly efficient multi-GPU sharding for large models in future work.
> * Table 1 is presented as the win rate of the preference tuned model output to the same prompt. We will make this more clear as a graph in the revision. We have also added GPT4 judge, TFLOPS, and time measurements as more detailed metrics. Please see the post above.
>
> Thank you and we look forward to your comments.
>
> **References**
>
> [1] Meng, Y., Xia, M. and Chen, D., Simpo: Simple preference optimization with a reference-free reward NeurIPS, 2024.
>
> [2] Angelica Chen, Sadhika Malladi, Lily H Zhang, Xinyi Chen, Qiuyi Zhang, Rajesh Ranganath, and Kyunghyun Cho., Preference learning algorithms do not learn preference rankings. NeurIPS, 2024.

---

### Official Review · Reviewer_aHcb · 2024-11-04

**Soundness:** 3
**Presentation:** 3
**Contribution:** 3
**Rating:** 5
**Confidence:** 3

**Summary:**

The authors have developed a  lightweight variant of Direct Preference Optimization (DPO) for aligning large language models with human preferences.  This is achieved by reformulating the DPO using a convex optimization reformulation of neural networks. Experimental results show the promise of the proposed approach.

**Strengths:**

Strengths of the paper include:
1. The paper addresses the scalability of DPO by offering a more efficient alternative
2. The use of convex optimization reformulation of neural networks as a surrogate for DPO loss appears to be novel.
3. The paper offers methods that could make DPO accessible to researchers without access to multiple GPU systems.

**Weaknesses:**

1. The proposed method is largely a combination of existing well-established approaches - convex function reformulation of neural networks, ADMM, JAX and hence the contributions are somewhat incremental.
2. There is little offered with respect to theoretical results on preference optimization
3. The experiments seem to be somewhat limited and do not include many benchmarks used by other researchers

**Questions:**

1. How do you expect your method to compare against SOTA DPO methods on other data sets such as the ones used in https://arxiv.org/abs/2407.13709, https://arxiv.org/pdf/2403.19159 and other related work?

---

> ### Author Response · Authors · 2024-12-04
>
> Thank you very much for your time and feedback! We have included citations to the works noted in the revision, and will discuss thoroughly.
>
> We have also added Section 3 which provides theoretical convergence guarantees for our work. Our benchmarks are replicated from the seminal work of the DPO paper, which also used 3 datasets on GPT4 judge and 25 human evaluators. We look forward to further discussions!

---

### Official Review · Reviewer_FPZ4 · 2024-11-08

**Soundness:** 1
**Presentation:** 1
**Contribution:** 2
**Rating:** 3
**Confidence:** 4

**Summary:**

This paper proposes a lightweight version of DPO that is supposed to require less resources.

**Strengths:**

It is desirable to make RLHF use less resources. Though note that this paper does not make DPO use less resources as was claimed in the abstract.

**Weaknesses:**

1. The introduction contains sweeping generalizations that are incorrect and not appropriate for an academic paper. For example, it is possible to perform RLHF on off-policy data. And it is hard to say "This model is able to infer what a human user wants and output a realistic answer that a human might like." This is not an accurate description of preference learning, and there are many documented issues with DPO and RLHF. Much of the writing throughout the paper is too informal and loose -- claims should be made more precisely and odd, extra text should be omitted. For example, much of the "JAX for Speed and Memory" section is written oddly (eg "our past work has found it to be extremely performant" with no citation, and "Recent research in review will provide more in depth discussion")

2. The emphasis on a 4090 GPU is a bit artificial -- a lot of the memory gain comes from tuning a smaller NN, not necessarily from the method. The authors should amend the introduction to mention the model size. This issue is even worse in Section 3, where the authors claim that their method circumvents the need for FSDP. Indeed, this is probably only the case because the model is smaller.

3. The experiments and results sections are clearly rushed. First of all, a win rate of "4" does not make any sense (Table 1). And speed and efficiency gains are claimed without any reported evidence. The evaluation in Section 4.4 does not match any standard notion of evaluation (yes, evaluation of generative models is hard, but if you want to say your method is better than another, you need to use some standardized evaluations). These are just some of the issues -- I am not listing them all.

**Questions:**

DPO should not cost any additional memory when implemented properly. One can run and save the logits of the reference model before starting training, and then preference tuning the model is just the cost of normal training. I am confused where the efficiency gain comes from besides tuning a smaller model, and I don't see any reported evidence besides informal gains ("roughly twice as fast", etc).


I honestly did not understand the method. Where does the two layer network even show up? How does adding an additional network result in an efficiency gain? What is this convex reformulation? Note that my lack of understanding for the method doesn't undermine my review, which was mostly focused on pointing out issues in the evaluation and experiments.

---

> ### Author Response · Authors · 2024-12-04
>
> Thank you for your valuable feedback! We realize the paper was initially unclear, and have explicitly updated the Algorithm with supporting theoretical proof.
>
> * DPO is not the same as RLHF. It is a more efficient alternative to RLHF, which can only be implemented by organizations with many resources. Despite offering improvements, DPO can still be very expensive. In this paper, we propose a new algorithm, CVX-DPO, as an alternative to DPO. CVX-DPO requires fewer resources to run, leading to convex optimization problems that are easy to solve on modern hardware and don't require extensive optimizer tuning to achieve good results. We apologize this was unclear in the original submission, but we believe the newly uploaded submission makes our contributions clear.
> * The introduction has been re-written to be more clear. The specific sentence you cited was quoted from one of the original authors of the seminal DPO paper during discussion about this work. We realize that although we respect the expertise of the original DPO authors, it is not professional to quote them verbatim in our introduction and have removed this. The paper has been rewritten to be more formal, and is no longer loose. We have deleted extra text.
> * We respectfully disagree with the reviewer on this point. When we say we can train on one 4090, we are talking about our method CVX-DPO relative to the original DPO, which runs into OOM memory issues even on these smaller models. For example, while CVX-DPO required one 4090 to train on the IMDb dataset with the GPT2-Medium model, we needed to use a cluster of A100s to get DPO to run. CXV-DPO also requires significantly less TFLOPS as seen above.
> * CVX-DPO's approach is model agnostic. The gains in memory do not come from using a smaller model but from the approach taken by CVX-DPO, which stacks a two-layer convex neural network on top of an existing pre-trained model. This results in an optimization problem that is much more computationally efficient to solve than the one in the original DPO, regardless of the size of the base pre-trained model.
> * Again, our comments on the need for FDSP concern DPO. While FDSP is needed for DPO to run on datasets such as IMDb, with CVX-DPO, we only needed one 4090.
> * The new metrics have been listed above, including somewhat standard evaluations, such as GPT4 judge feedback which was used in the original DPO paper. The choice of 25 human samples was also implemented to match the seminal DPO experiments.
>
> We do agree that for the largest models, sharding across multiple devices will be greatly advantages. Especially since our JAX codebase readily admits to highly efficient sharding in future work. In summary, CVX-DPO will still require fewer resources than DPO due to the nature of its formulation. Thank you for your feedback!

---

### Note · Authors · 2025-01-12

I have read and agree with the venue's withdrawal policy on behalf of myself and my co-authors.